# Analysis of Risk Factors in Patients with Subclinical Atherosclerosis and Increased Cardiovascular Risk Using Factor Analysis

**DOI:** 10.3390/diagnostics11071284

**Published:** 2021-07-16

**Authors:** Zuzana Pella, Dominik Pella, Ján Paralič, Jakub Ivan Vanko, Ján Fedačko

**Affiliations:** 1Department of Cybernetics and Artificial Intelligence, Faculty of Electrical Engineering and Informatics, Technical University of Košice, 040 01 Košice, Slovakia; jan.paralic@tuke.sk (J.P.); jakub.ivan.vanko@student.tuke.sk (J.I.V.); 21st Department of Cardiology, East Slovak Institute for Cardiovascular Diseases, 040 01 Košice, Slovakia; dominik.pella@gmail.com; 3Faculty of Medicine, Pavol Jozef Šafárik University in Košice, 040 01 Košice, Slovakia; 4Centre of Clinical and Preclinical Research, MEDIPARK, Faculty of Medicine, Pavol Jozef Šafárik University in Košice, 040 01 Košice, Slovakia; jan.fedacko@upjs.sk; 5Department of Gerontology and Geriatrics, Faculty of Medicine, Pavol Jozef Šafárik University in Košice, 040 01 Košice, Slovakia

**Keywords:** cardiovascular diseases, factor analysis, risk factors

## Abstract

Today, there are many parameters used for cardiovascular risk quantification and to identify many of the high-risk subjects; however, many of them do not reflect reality. Modern personalized medicine is the key to fast and effective diagnostics and treatment of cardiovascular diseases. One step towards this goal is a better understanding of connections between numerous risk factors. We used Factor analysis to identify a suitable number of factors on observed data about patients hospitalized in the East Slovak Institute of Cardiovascular Diseases in Košice. The data describes 808 participants cross-identifying symptomatic and coronarography resulting characteristics. We created several clusters of factors. The most significant cluster of factors identified six factors: basic characteristics of the patient; renal parameters and fibrinogen; family predisposition to CVD; personal history of CVD; lifestyle of the patient; and echo and ECG examination results. The factor analysis results confirmed the known findings and recommendations related to CVD. The derivation of new facts concerning the risk factors of CVD will be of interest to further research, focusing, among other things, on explanatory methods.

## 1. Introduction

More than 4 million people die each year in Europe from cardiovascular disease (CVD). Despite the progress in CVD treatment, it will not be significantly reduced by the prevalence of coronary heart disease. The cause of this condition is either unrealized or late indicated diagnostic methods for early diagnosis of coronary artery disease. Modern drug therapy and specialized intervention centers are helpful for the prevention and treatment of CVD. However, coronary artery disease and preventive medicine screening could be better, confirming data from Euroaspire I-V programs. We have modern PCI (percutaneous intervention) centers, but patient’s medical care after hospitalization has some limitations [1,2,3]. The timeline of the clinical trials and the available data are only the traditional risk factors included in this scoring system. Consequently, the system lacks certain crucial aspects, including, among other things, the presence of diabetes mellitus type 2, smoking, obesity, or the results of the newest clinical trials dealing with the inflammatory theory of atherosclerosis, genetic parameters, and respectively familiar hypercholesterolemia differences. Waiting for invasive diagnostic and therapeutic methods is often the subject of delay, caused by the inadequate selection of patients undergoing selective coronarographies due to the unnecessary examinations of low-probability patients. This over-diagnosing is not associated with a better prognosis; it merely highlights the need to optimize the examination process, especially personalized medicine, artificial intelligence (AI), machine learning (ML), or statistical methods. These personalized aspects based on the patient and the available data concerning the cardiovascular risk and the risk stratification of positive coronarography findings could bring us more effective diagnostics and treatment of cardiovascular disease [4].

Artificial intelligence can help in some aspects of medical care, patient management, and examination (better diagnostic, decreased risk of iatrogenic complications, better recovery). A set of methods is used for this, including ML methods. The growing importance of ML methods lies in the fact that these methods can process vast volumes of different types of data in a relatively short time. Likewise, the number of such research studies in medicine is constantly growing, as medicine is extremely rich in data. In addition, ML offers different options depending on the methods used. The most typical example is the classification of patients according to the presence of a particular disease. However, there are other perspectives on medical issues. As mentioned above, risk factors lead to disease, and early detection of risk factors can help diagnose a disease in its early stages. It can lead to specific changes that can subsequently improve the patient’s condition. Here we come to the concept of personalized medicine.

However, for the enormous amount of data, providing personalized medicine is a challenging task. We come to the potential of ML methods, respectively AI, which also includes ML. Patient data contained in electronic health records (EHR) must first be extracted [5], then data can be used to train algorithms that are part of Clinical Decision Support Systems (CDSS) [6]. Such systems then generate recommendations based on a combination of available theoretical knowledge and facts “read” from the analyzed data. However, as stated in [6,7], these recommendations should not be taken in any other way than the recommendations. They should not be used as automated diagnostic procedures, as they cannot replace a trained healthcare professional.

The presented work aim is to provide a less typical view of the issue of risk factors. Specifically, our effort will be to examine the existence of clusters between the attributes of the monitored dataset. For this purpose, we focus on Factor Analysis, revealing the clusters of the observed characteristics. We subsequently extended our decision support system for cardiologists (DSSC) (not the presented work’s subject).

The rest of this paper is organized as follows. Section 2 presents our analysis of related work motivating our research in this direction. Section 3 describes the background of the data collection, the data used for experiments, and the necessary preprocessing operations. Section 4 presents details about the used research method-factor analysis and the results of our experiments. Section 5 discusses the results achieved in a broader context and their implications on medical practice. Finally, Section 6 concludes this paper and sketches the future directions of our research.

## 2. Related Work

In recent years, the use of factor analysis, not only in the field of CVD, has been increasing. Tsai and co-workers [8] focused on the relationship between cardiometabolic factors, systemic inflammation, and data on the Taiwanese population’s sitting time and physical activity using principal component analysis. They created five clusters of factors and demonstrated that traditional CVD risk factors are not similar to those describing the physical activity that falls into one cluster. The combination of factors they created was able to explain almost 70% of the variability of the 14 examined characteristics. Another significant result of their work described the interrelationship between inflammation and adiposity. In conclusion, they also suggest that insulin resistance and inflammation are signs of adiposity.

The connection between factor analysis and CVD was grasped by Tsai’s research [9] in terms of modifiable risk factors within the Taiwanese population. In this way, they created five factors corresponding to the usual guidelines. The analysis showed an intense significance of waist circumference, blood pressure, and total cholesterol related to CVD. Their results indicated that the prevalence of metabolic syndrome in Taiwan was higher in men (15.5%), but found no significant gender difference compared to other CVD risk factors. In conclusion, the research authors evaluated that their research revealed other risk factors for CVD, including lifestyle factors, exercise, and total cholesterol. They point out that early identification of modifiable risk factors may play an essential role in preventing CVD.

Both studies focused on the middle-aged to elderly population. However, as [10] points out, the atherosclerotic process begins much earlier, so they decided to target the child population (8–10 years old) in New Zealand. The study’s authors identified four factors describing blood pressure, adiposity, lipids, and vascular, which explained the 60% variance of the monitored variables. The blood pressure factor explained almost 40% of this variance. They also confirmed that glycated hemoglobin and fibrinogen acted on several factors, and fibrinogen suggests short-term glycemic control. They identified a higher risk of adiposity, vascular and cumulative risk score in overweight and obese children.

Another study looking at the association between factors of hospital practice related to heart failure and mortality at 7 and 30 days after hospitalization and during hospitalization in Japanese hospitals, respectively, provided a different perspective [11]. The researchers identified five factors describing interventional cardiology (F1), cardiovascular surgery (F2), pediatric cardiology (F3), electrophysiology (F4), and cardiac rehabilitation (F5). This combination of factors explained approximately 100% of the variance of the original data (90 observed characteristics), with the factor of cardiovascular surgery contributing the most. Subsequently, the association was evaluated based on logistic regression. As a result of their research, while F3, F4, and F5 were associated with lower mortality during the period under review, factor F1 was associated with a higher mortality, and factor F2 did not show any significant association with mortality.

As we have pointed out, factor analysis can be used in one domain, the CVD domain, from different angles. It is possible to focus on risk factors either in the adult population or in the child population. It is possible to focus on a specific group of patient characteristics obtained from healthcare records or questionnaires. It is possible to focus on medical procedures in connection with the continuation of the disease. Most of the research was carried out in Asian countries or America. Studies linked to the European population are not significantly represented. Another gap is also shown in the direction of factor analysis in the field of CVD. In the survey, we did not find any similar study that would focus on commonly monitored patient symptoms obtained by laboratory tests or admission examinations following coronary heart disease and coronary artery obstruction.

## 3. Background of the Data Collection, Overview of Data and Data Preprocessing

The idea of the mentioned-above decision support system for cardiologists was based on collaboration between the East Slovak Institute for Cardiovascular Diseases in Košice (ESICD), the Faculty of Medicine of the Pavel Jozef Šafárik University in Košice (FM PJŠU), and the Department of Cybernetics and Artificial Intelligence of the Faculty of Electrical Engineering and Informatics of the Technical University of Košice (DCAI). A common multidisciplinary research team is developing DSS within the Kosice Selective Coronarography Multiple Risk (KSC MR) Study project. The idea of KSC MR is to use ML methods to analyze the characteristics of patients, including physical findings, laboratory examination, questionnaires, and, among other things, the results of selective coronarography. All the significant results from the processing and modeling of these data should be part of the software (in the form of DSSC), which will enable the early detection of high-risk patients concerning the possibilities of personalized medicine [12]. The KSC MR study plan was to enroll prospectively approximately over 1000 patients aged 18 and above, without an upper limit, who will be admitted to the associated hospital suspected of having coronary artery disease.

The population of our study was collected on the 1st Cardiology department of ESICD. We randomized patients with one of the more known risk factors of CVD and subclinical atherosclerosis detected by CT angiography or other noninvasive imaging or exercise examinations. Every patient from our study has performed selective coronarography examination (diagnostic, and if also needed, therapeutic), assessment of laboratory parameters. The exclusion criteria were—known coronary artery disease and history of acute coronary syndrome.

Patients were hospitalized in ESICD between June 2017 and March 2018. Data collection was performed retrospectively concerning the described inclusion and exclusion criteria. Electronic health records (EHR) underwent natural language processing and were further analyzed as a set of structured data [5]. All experiments and work with the data file were performed in the R Studio environment (version 1.4.1106) in the R programming language (version R-4.0.2).

Overall, the data contains 66 attributes, which we divided into the following groups: identifying attributes, symptomatic attributes, and resulting attributes. Each group of features is capturing specific information about patients. An overview of the used attributes and their essential characteristics and distribution of values is described in Table 1.

The degree of severity of the finding of coronarography was derived from the narrowing of the coronary arteries of the heart found during the examination of selective coronary angiography as follows:0—no finding;1—at least one branch contains 10% narrowing;2—any branch, except the RIA basin, has 20 to 50% stenosis;3—RIA branches have 20 to 50% narrowing, the other 50 to 70% narrowing;4—RIA branches have a maximum of 70% narrowing (50–70%), the other 70–100%;5—at least one of the RIA branches has more than 70%.

The distribution of values of the Coron_result attribute is presented in the following Figure 1.

### Data Preprocessing

Although the data contain a relatively large number of attributes describing the patient from different angles, it was necessary to adjust them for our needs. The first step was to infer the patients’ age (using the Year (year of birth of the patient) and ID attributes which contain information about the patient’s year of hospitalization). Therefore, we got the attribute Age, which tells us that our cohort’s average age was 65.23 (min: 36, sd: 9.34, median: 66, max: 88, IQR: 13).

Looking at the data structure and distribution of the values shown in Table 1, it can be seen that not all values are correct. Incorrect entries can have their origin in patient records or result from an error in the processing of medical reports. We have described this process in more detail in the work of [5]. One of the problems can be observed on the extreme values, outliers, which appear in the BMI (body mass index) attribute. Since the calculation of the BMI value depends on the Height and Weight attributes, we must look at these attributes together. In Figure 2a, the distribution of values displayed using the so-called boxplot suggests the existence of outliers. The reader may notice a change in the distribution of the values by comparing boxplots (a) and (b) of Figure 2.

The second problem with outliers occurred with the BP (blood pressure) attribute. The minimum value for this attribute was 12, and the maximum was 14,090. Both values represent extremes that are incompatible with human life. We replaced the extreme values by the 5th or the 95th percentile of a given attribute. The result can be seen in Figure 3, which indicates that the range of values of the BP attribute is within the normal range.

The cardiologist’s recommendation was to add another essential and well-known attribute, the ESC risk score, representing the rate of cardiovascular risk for the nearest ten years [13]. We created this attribute using several features from our dataset: the age of patients, gender, cholesterol, and blood pressure. The average value of ESC was 7.45 (min: 0, sd: 5.06, median: 6, max: 33, IQR: 5). Since many attributes contain NA values, this was also reflected in ESC attributes, and thus ESC has 431 NA values.

The next step in preparing the dataset was to treat attributes that acquired only one type of value, which can be a problem for Factor Analysis. Specifically, these were the attributes R_Hyperch (occurrence of high blood pressure in the patient’s family history), HIV (outcome of the patient’s HIV test), HBsAG (presence of hepatitis B virus antigen in the patient), and ECG_LBBB (presence of left Tawar shoulder block in the patient). Therefore, we could not use these attributes, and we decided to remove them from the dataset.

Another problem was that data contains 4749 missing values. In Figure 4, the reader can see the distribution of missing values in %. As shown in Figure 4, the most considerable amount of missing values is in the Chloride attribute, almost 80%. The FBG (fibrinogen levels) attribute follows it with approximately 77% missing values.

Based on the analysis of algorithms for the imputation of missing values [14], we decided to focus on more sophisticated methods. We used a known method with reported good results, k Nearest Neighbors (kNN), filling in the missing values through single imputation.

The last step of the preprocessing data phase was to deal with the attributes describing coronary angiography. In the first part of this step, we removed all the attributes describing the results of coronary angiography, based on which the attribute Nalez was also created. Specifically, the following attributes, represented the percentage narrowing of the heart arteries, were removed: ACS, RIA, RD1, RD2, RCX, RIM, RMS1, RMS2, ACD, and RIP, as well as the Muscle_bridge attribute.

Subsequently, we removed this target attribute to prevent it from being selected as part of one of the factors created during the implementation of the Factor Analysis. In addition, our goal was to find connections between features, ideally without considering the resulting attributes derived from coronary angiography.

Part of the data preprocessing was also the transformation of all attributes to the numeric data type.

## 4. Methods—Factor Analysis

Factor analysis (FA) is a statistical method that ranks among multivariate exploratory techniques, and its origins date back to 1904 [15]. The source of its roots is attributed to Charles Edward Spearman. Initially, the FA focused on its use in psychology, and only later did it find application in other fields.

As F. Gorunescu writes [16], multivariate exploratory techniques focus on finding hidden patterns in multidimensional data. He further states that FA can be used in two ways:Reduction of the number of attributes to reduce the computational time in data processing;Detection of the structure of connections hidden in the data.

Using FA on data is a reduced/transformed set of data in both cases, where interrelated attributes created, respectively, are replaced by new variables called factors. 

The pair of authors Olson and Lauhoff even state in [17] that FA can be used as a preprocessing technique to get an idea of the number of clusters and review the dataset’s interrelated features. Another pair of authors, Wang and Kuo [18], agree with this view and describe FA as an essential step towards effective aggregation and classification procedures. They further add that the principal goal of FA is to find and rank crucial factors that can sufficiently represent the problem to be solved. A suitable number of factors needs to be chosen, and they should have two fundamental characteristics: independence and importance. These properties can be considered a necessary condition for FA. Wang and Kuo also point to a sufficient condition of FA, which indicates that the factors should be able to represent complete information about the system, which the amount of undiscovered knowledge can measure.

### 4.1. Evaluation of the Suitability of Using FA on a Selected Dataset

Before we get to the factor creation phase, it is necessary to verify whether the given dataset is suitable for the FA application. For this purpose, we will focus on two statistics in our work. The first is the Kaiser-Mayer-Olkin measure (KMO), which evaluates the degree of homogeneity of the observed features and is considered the overall measure of the adequacy of the experimental dataset. For KMO values, the Kaiser and Rice recommendation (see also Table 2) is used in practice [19,20].

The second statistic observed is the Measure of Sampling Adequacy (MSA), which indicates the extent to which others predict individual attributes. In other words, it evaluates the degree of correlation between unique characteristics [21]. As with KMO, if the MSA is below 0.5, the investigated dataset is unsuitable for FA use. The results of this calculation for our dataset can be seen in Figure 5, where the KMO (Overall MSA) rate is 0.61, corresponding to the average suitability of using FA on our dataset. The MSA rate is listed separately for each attribute, and although not all attributes take values above 0.5, we chose to keep them.

### 4.2. Determination of the Approximate Number of Factors

As described in [21], it is further necessary to determine the appropriate number of factors. Here, too, there are several approaches to determine the most appropriate number. Since [22] recommends using several methods to select the optimal number of factors, we focused on the Kaiser criterion, Scree test, and Parallel analysis. 

#### 4.2.1. Scree Test

The Scree test helps determine the approximate number of factors suitable for a given dataset based on the point where the course of the Scree graph significantly changes (in the so-called elbow). The horizontal line y = 1 in the Scree plot determines the limit to how many factors whose eigenvalue is greater than 1. The Scree plot is often referred to as a rubble test. In Figure 6, we can notice two waveforms of the graph. Both are important for us, i.e., both the intrinsic values of the factors and the eigenvalues of the components [23]. The first line of the graph (corresponding to FA) determines the eigenvalues of the factors. Based on the Scree plot recommendation, we should focus on creating three factors. The second line of the graph corresponds to the actual values of the components (PC), while it is recommended to develop 19 components.

#### 4.2.2. Kaiser Criterion

Since the course of the Scree plot may not always show an unambiguous result, another researcher came up with a proposal on how to estimate the number of factors. Hanry F. Kaiser [24] proposed Kaiser’s rule, which directly examines the intrinsic values of factors. According to this rule, only factors whose eigenvalues are above one are retained. However, ref. [25] indicates that the number of factors with eigenvalues greater than 1 is related to the number of variables examined and is typically in the range of 1/4 to 1/3 of the number of variables. Ref. [25] refers to simulation studies that show that the Kaiser criterion is sometimes inaccurate; determining the number of factors in this way is not considered reasonable. However, we looked at this rule out of interest. The result is two recommendations:Empirical Kaiser Criterion suggests 23 factors.Traditional Kaiser Criterion suggests 19 factors.

#### 4.2.3. Parallel Analysis

Horn introduced parallel analysis (PA) in 1965, and its original purpose was to improve the Kaiser criterion [25]. It also extends the rubble test. In addition to the essential display of the Scree graph, it also generates random correlation matrices, which are the basis for the averages of the eigenvalues of the random data displayed in the chart. Components, respectively factors, are retained if the intrinsic value from the actual data exceeds the intrinsic value from the random data [23,25]. Figure 7a shows graphs for estimating the number of factors (or components) using PA. The blue line is the same as in the Scree graph and corresponds to ‘FA’. However, we do not observe when it intersects with the horizontal prime y = 1, but we keep its intersection with a red line. In estimating the number of factors (the course corresponding to FA in Figure 7a), we get to a higher number than the recommendation using a rubble test. In this case, the recommended number of factors is 15.

Within R language packs, we can perform PA in another way, too. The result is shown in Figure 7b. In this case, we compare the course of the blue and red lines. These corresponds to the intrinsic values of the factors (red observed, blue random). Again, the number of significant factors can be estimated based on their intersection. In our case, the considerable number of factors is equal to 23.

Goldberg and Velicer [25] again refer to simulation studies that find that parallel analysis is one of the most accurate methods. In our case, the PA confirmed the conclusions of the application of the Kaiser criterion.

Based on the performed analyses, we decided to create several variants for 3, 15, 19, and 23 factors.

### 4.3. Factor Rotation

The next step of the FA was to select the appropriate rotation type of the factor axes. Factor rotation was presented shortly after the discovery of FA. This performance of factor rotation has led to an effort to better explain and interpret the FA results. The term rotation comes from the fact that the axes are rotated so that the clusters of factors were as close as possible. Axis rotation methods can be divided into orthogonal and oblique (for the angle between the x and y axes). The fundamental difference between these groups of techniques is that while oblique rotations create correlated factors, orthogonal rotations create uncorrelated factors. Thus, orthogonal rotations provided a more uncomplicated explanation of the factors. However, there are areas of research where it is challenging to expect factors to be uncorrelated. In this case, the use of orthogonal rotation may lead to a less helpful solution. Conversely, using shim rotation is unlikely to skew the results if the factors are uncorrelated. For each rotation method, the criterion of simple structure is considered, and thus it is required that most factor loadings be high for only one factor, and for the others, they should be around zero [25,26].

The most used method is Varimax, a type of orthogonal rotation. Its wide application minimizes the number of variables with high loadings and thus simplifies the interpretation of factors. Other examples of orthogonal rotation methods are Quartimax (aimed at minimizing the number of factors) and Equamax (reduces the number of variables with high loadings and the number of factors needed to explain the individual variables). Of the group of oblique rotation methods, two are most used: *Direct oblimin* and *Promax*. The fundamental difference between these methods is the speed of their calculations. *Promax* is faster than Oblimin and is, therefore, the recommended method of oblique rotation for large volumes of data [25,27]. In, ref. [28] choosing Varimax and Oblimin to begin with is recommended and we will stick to them in this work.

### 4.4. Factor Analysis Modeling

After verifying the suitability of the dataset for FA, selecting the recommended number of factors (3, 15, 19, and 23), and choosing the axis rotations (Varimax and Oblimin), we could proceed to FA modeling. In total, we created eight models, but we will present only the most important ones.

First, we will deal with the orthogonal rotation of factors, Varimax. We performed separate modeling for each number of factors. The 15-factor model already showed signs of excessive factor extraction, as the two factors consisted of only one-factor loading. Nevertheless, we also looked at the remaining planned models. As we expected, the number of factors with one-factor loading increased; Smyth and Johnson [23] also draw attention to excessive extraction, where the research authors subsequently extract a smaller number of factors. The authors in [22] also state that at least two or three-factor loads must load a factor to give meaningful results. Thus, we also reduced the number of factors, and the first modeling result, which did not show signs of over-extraction, was for ten factors. Figure 8a shows a model for three factors and Figure 8b for ten factors.

Subsequently, we focused on the rotation of Oblimin factors. Similarly, it was shown that over-extraction already took place during the formation of 15 factors. As with the Varimax rotation, we proceeded to reduce the intended number of factors. The first result, when the factor was not formed by only one-factor loading, was obtained by creating ten factors. The results can be seen in the following Figure 9.

All selected factor loads reached at least 0.3 in absolute terms. According to [29], a factor load value above 0.3 can be considered significant, although this may also depend on the data sample size. For better clarity, we also present a table of individual solutions—Table 3.

Let us analyze the result of all models in more detail. We will first focus on the value of SS loadings, which represents the sum of squared loadings. If a given factor acquires this value above one, the given factor should have a specific informative value and should be maintained [30]. The following Table 4 shows the SS loading values for each FA model.

As the reader can see, most of the values are above the recommended acceptance limit. However, in the case of the ten-factor model, two factors do not reach the SS loadings threshold above level one for both factor axis rotations. For the sake of interest, we also added the result of observing the values of SS loadings for a model consisting of 23 factors. In this case, only seven factors reached the value of SS loadings above one; the rest were below this limit (rotation ‘Varimax’). To maintain the condition of SS loadings > 1, we proceeded to reduce the number of factors.

The first usable result for the value of SS loadings was obtained by creating eight factors. The following Figure 10 is a representation of these models.

For better clarity, we presented the following Table 5 of factor loadings and SS loadings for individual factors.

The second thing we will look at in the created models is the cumulative variance of each n-factor model. This value indicates how many (in %) of the monitored attributes a given n-factor model can explain. If these values are small, a more significant number of factors must be selected [30]. Again, we offer a report in the form of a table.

As can be seen from Table 6, the variance of the models we created is relatively small (only 13% for the three-factor model and 26% for the eight-factor model), which means that these models do not meet the conditions discussed above.

Let us look at the composition of individual factors in terms of their medical significance. We will focus only on some unique models, as the total number of factors is not negligible, and the models are pretty similar.

As in the first model, we will choose a three-factor model based on Varimax rotation. The broader context of cardiovascular risk and rough groupings of related attributes is evident for this model.

Factor 2 (F_CAD; F_MI; Urea; ECHO_EF; ECHO_PH; ECG_Rhythm) shows us some genetic background of cardiovascular diseases. There is a typical connection between family history of coronary artery disease and status after myocardial infarction and patient parameters from examinations: changed renal parameters, ECG changes, and systolic function of left ventricular of the hearth. These are well-known data supported by many great population studies [1,2,3]. 

Factor 3 (P_MI; P_CAD; P_DM; P_Stroke) shows a prevalent connection with cardiovascular risk and traditional risk factors. The patient who has a personal history of diabetes and status after stroke also has a connection with coronary artery disease and myocardial infarction. This is a known knowledge about cardiovascular risk factors [1]. Factor analysis shows us non-inferiority and detected connections between coronary artery diseases and their risk factors.

The model for the Oblimin rotation is very similar. The difference is only in the location of two-factor loadings—Age, in the case of Varimax rotation, loads Factor 1 and thus frees up the place of Weight; in case of Oblimin rotation, Age loads Factor 2.

Another solution we will look at is the ten-factor rotation model Oblimin. Again, it is very similar to the result of the Varimax rotation; the only slight difference is in the significance of the individual factors (SS loadings value) and the inclusion of the ECHO_PH factor load Varimax rotation results.

Factor 2 (Urea; Creat; FBG) shows us the typical connection between renal parameters and fibrinogen levels. This connection could have an inflammatory background, increasing renal parameters and fibrinogen levels. There is a place for advanced research in the next steps of our study [31].

Factor 3 (F_CAD; F_MI) shows us that if the patient has in family history suspicion of coronary artery disease, status after possible acute coronary syndrome (myocardial infarction) or stroke, then there is an increased risk of cardiovascular disease [1].

Factor 4 (P_MI; P_CAD; P_Stroke) shows us the frequency of coronary artery disease, and if this frequency increases, then there is also an increased frequency of acute coronary syndromes, especially myocardial infarction [1]. Factor 7 (ESC; BP Age) shows us the typical connection between cardiovascular risk and age and blood pressure. Increased blood pressure and age are traditional risk factors of coronary artery diseases [1].

As for Factor 8 (ECG_FR; ECG_QT) and Factor 10 (F_DM; F_HT), they are likely to have some significance (albeit very small) and their low informative value. This fact is also confirmed by the FA conclusions, where the SS loading value for these factors was below level one. 

Lastly, we will discuss the eight-factor solution. As indicated by the value of SS loadings for a ten-factor solution, when the number of factors was reduced to eight, two factors below the required limit were wholly omitted from the solution. Using the rotation Oblimin, the structure of factor loads of individual factors did not change at all; there was only a slight change of order for the values of SS loadings. The Varimax rotation also changed the arrangement of the factors according to the SS Loadings value and shifted or supplemented the factor loads ECHO_PH (for the ten-factor model loaded by Factor 8) ECG_Rhythm of loading Factor 2.

The last series of experiments we performed involved a slightly modified dataset. Returning to Figure 5, MSA statistics for individual attributes should be above the level 0.5 [20]. The data file we worked with also contained the monitored characteristics below this level—specifically BP, AST, Sodium, Chol, ECG_SVES, ECG_STD. We therefore proceeded to remove these attributes from the dataset.

Again, we performed analyses on such an adjusted set to determine the appropriate number of factors. The values recommended by particular analyses methods were as follows:Scree plot: 3 factors;Kaiser criterion: 17 factors;Parallel analysis: 11 factors.

When creating individual clusters of factor loads, a similar result was crystallized as in the previous phases. The solution of ten factors did not reach the value of SS loadings above level 1 for all factors. We achieved this result only when reduced to six factors for both the Varimax rotation and the Oblimin rotation. As we expected that the characteristics should correlate with each other due to the nature of the monitored data, in the following section, we will only approach the solution of the rotation Oblimin (Figure 11, Table 7).

The factors obtained can be identified by the following descriptions: Factor 1 describes the ‘Basic characteristics of the patient’. As several recommendations state, the difference in cardiovascular risk for men and women is that the patient’s height and weight correspond to the BMI (body mass index). Factor 2 can be characterized as ‘Renal parameters and fibrinogen’. In order, Factor 3 focuses on the ‘Family predisposition to CVD’. Factor 4 aggregates the loads describing the ‘Personal History of CVD’. The acquired Factor 5 points to a kind of ‘Patient’s lifestyle’ regarding bad habits. The last Factor 6 created describes ‘Echo and ECG examination results’.

Regarding the degree of variance for individual factors themselves, we can state that they are very equivalent. However, the degree of variability values is relatively low, in the order of 4.7%, 3.6%, 4.2%, 4.2%, 4.2%, and 2.9%. The following Table 8 shows the values of individual factor loadings for each of the created factors.

### 4.5. Interpretation of Factor Analysis Results

Although we focused on all the recommended procedures regarding the correct selection of the examined attributes, selection of the number of factors, considering the actual values of factors in the form of SS loadings, the nature of our data shows a relatively low variance for any cluster of factor loads. In the last model we presented (the six factors with rotation *Oblimin*), the achieved variance rate of 23% does not stand out from the other proposed solutions. However, the experiment results provide many fascinating known cardiovascular risks and cardiovascular disease axioms. The result of factor analysis confirm many well-known connections in cardiology. This shows us the very good potential and consequences of factor analysis, and our results confirm the conclusions of many randomized or population cardiovascular clinical trials [1,2,3,32,33].

### 4.6. Limitations of Our Study

We collected over 800 patients’ data. This could limit the smaller patient clusters if we compare it to significant population studies. All laboratory parameters were not collected at the same medical laboratory. There could be minor differences in the sensitivity and ranges across the used medical laboratories. 

Another limitation may be the number of factors chosen. Although we have used several methods to estimate the most appropriate number of factors, it is still a subjective choice. Likewise, the interpretation of the obtained factors can be considered subjective, even though we can substantiate the found connections with the performed research and professional literature.

## 5. Discussion

The present study describes the use of the factor analysis method on the data of patients hospitalized at the East Slovak Institute for Cardiovascular Diseases in the period between June 2017 and March 2018 to perform a selective coronary angiography examination. The examined data file contains 808 records and 66 monitored characteristics. The FA method identified six factors: (1) Basic characteristics of the patient; (2) Renal parameters and fibrinogen; (3) Family predisposition to CVD; (4) Personal history of CVD; (5) Lifestyle of the patient; and (6) Echo and ECG examination results.

According to the value of SS loadings (sum of squared loadings), the essential Factor 1, named as Basic characteristics of the patient, describes a person’s essential characteristics (gender, height, and weight), while the individual monitored characteristics of gender and height reached relatively high values of loads. The patient’s gender is one of the risk factors, as men have a higher CVD risk. The conclusion that the patient’s gender is a significant risk factor is also underlined by the result of the study [9], where the authors drew different results for men and women. The patient’s height itself is one of the indicators of BMI that indicates patient obesity. The patient’s weight is also related to another, often-observed phenomenon, the circumference of the patient’s belt. Several research results [8,9,10,34,35,36] are fully or particularly associated with these characteristics.

The correlation between the possible inflammatory effect of elevated fibrinogen levels and renal parameters was named Renal parameters and fibrinogen. The markers of inflammation and the plasma levels of IL-1β, IL-1RA, IL-6, TNF-α, hs-CRP and fibrinogen were higher among participants with lower estimated glomerular filtration rates. Inflammation score was higher among those with lower eGFRs and higher Urine Albumin-to-Creatinine Ratios (UACR). Biomarkers of inflammation were inversely associated with measures of kidney function. Atherosclerosis also has inflammatory pathogenesis, and patients with acute or chronic renal diseases have increased cardiovascular risk [37]. Fibrinogen also appeared in some of the studies analyzed, either as part of the adiposity [38] or in connection with smoking [39]. For example, urea was also represented in the research as a load on the metabolic factor [8,9]. 

Another critical factor is the CVD Family Predisposition Factor, which relates to coronary heart disease and MI attributes. Very few studies would independently identify them as significant factors corresponding to any family predisposition to CVD. However, Marušič’s study [38] also included a family history of coronary heart disease in the FA, resulting in a relatively high load factor. Collingwood also considered the family history of CVD [40]. Several studies have focused on a group of patients who had some CVD or related diseases, e.g., for the use of FA for patients suffering from DM [41], a group of patients affected by IM [40,42].

The analysis’s focus in [41,42] indicates that the personal history of CVD itself is significant concerning the further course of CVD. The positive family history of CVD itself is evaluated based on recommendations with a positive correlation to the patient’s predisposition to CVD. 

Factor 5 describing the patient’s lifestyle has appeared regularly in several studies, particularly smoking and alcohol consumption [9,38,40,43]. These conclusions also agreed with several expert recommendations that make smoking one of the risk factors associated with CVD [1,2,3,32,33].

The data from population clinical trials MESA or the Framingham study reflected many relationships between noninvasive or imaging examination results. Our factor analyses show a significant correlation between the ejection fraction of the left ventricular of the hearth measured by trans thoracal echocardiography, age, and ECG changes (QRS complex, morphology, duration, and right bundle branch block (RBBB)). There are no relevant data (clinical trials with a large population) to confirm this relationship. Some clinical trials examined the relationship between the systolic function of the left atrium and RBBB in patients after MI [44]. 

The presented results show us the interesting potential of factor analysis. There is a place for implication for making decisions about new researches based on molecular medicine, proteomics, or mRNA screening. This method could be a ‘gatekeeper’ for better orientations between complex connections and various factors in medicine.

## 6. Conclusions

The results of the factor analysis confirmed the known conclusions and recommendations related to CVD. The derivation of new facts concerning the risk factors of CVD will be of interest in further research focusing, among other things, on explanatory methods. The indicated relationships between CVD risk factors can be investigated using different machine learning methods, such as association rules, regression analysis, or a visualization tool for Bayesian networks.

## Figures and Tables

**Figure 1 diagnostics-11-01284-f001:**
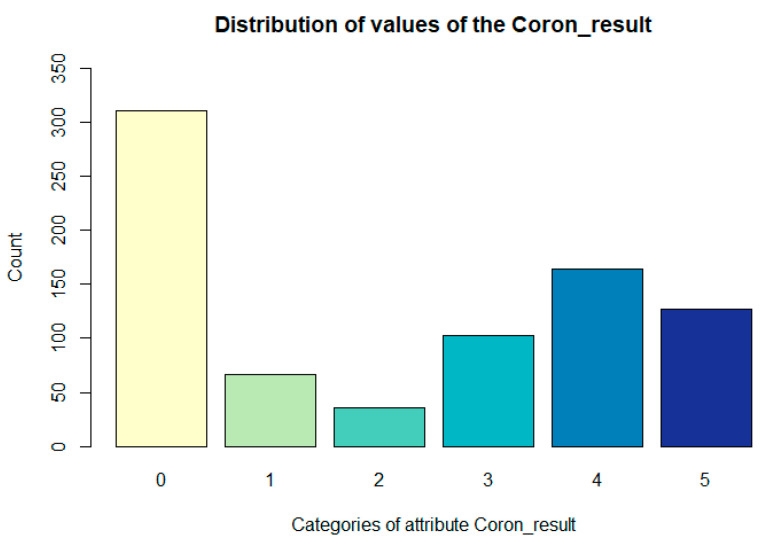
Representation of values Coron_result (the degree of severity of the coronarography finding) attribute.

**Figure 2 diagnostics-11-01284-f002:**
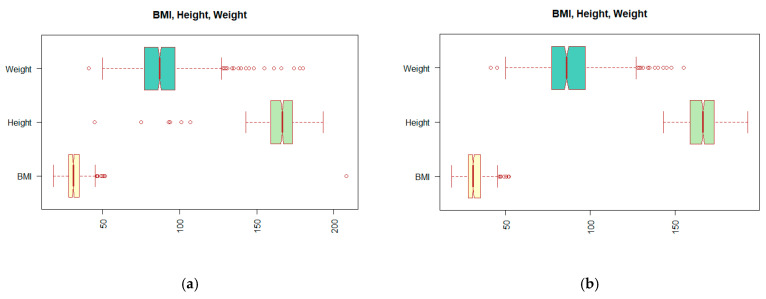
Display boxplots for attributes BMI (body mass index), Height, and Weight: (**a**) before outlier treatment; (**b**) after treatment of the outliers.

**Figure 3 diagnostics-11-01284-f003:**
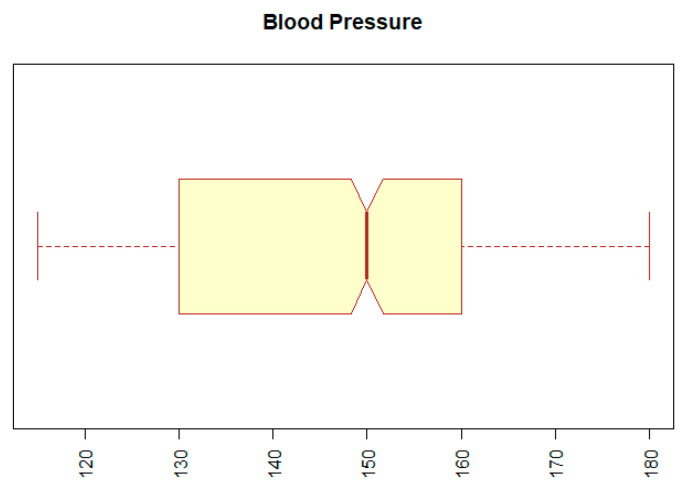
Boxplot for attribute BP (blood pressure).

**Figure 4 diagnostics-11-01284-f004:**
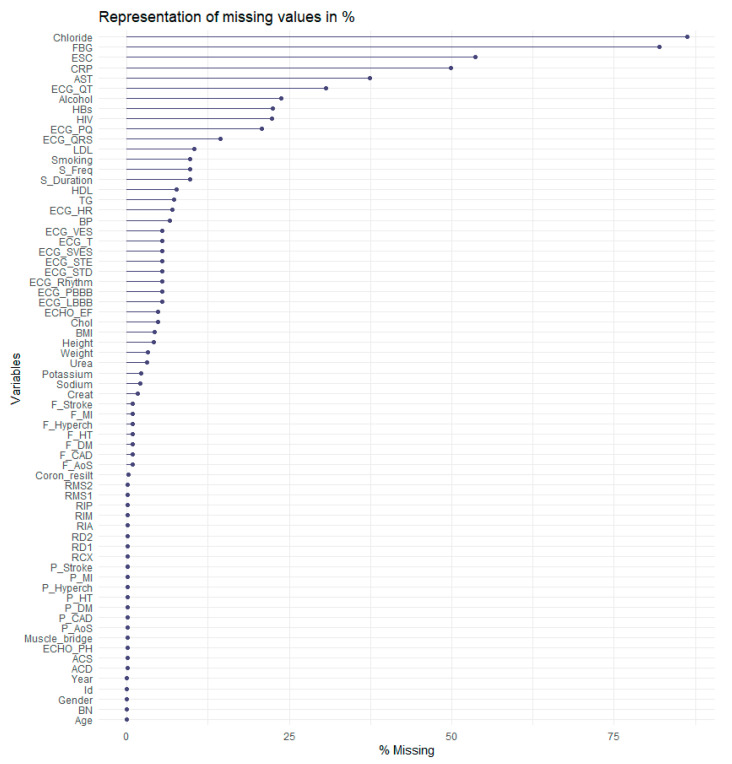
Representation of missing values.

**Figure 5 diagnostics-11-01284-f005:**
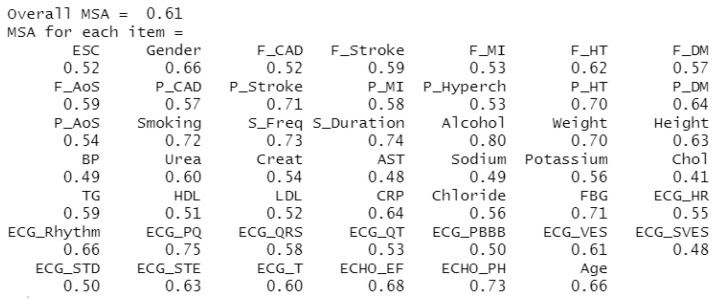
Representation of KMO and MSA values for monitored attributes.

**Figure 6 diagnostics-11-01284-f006:**
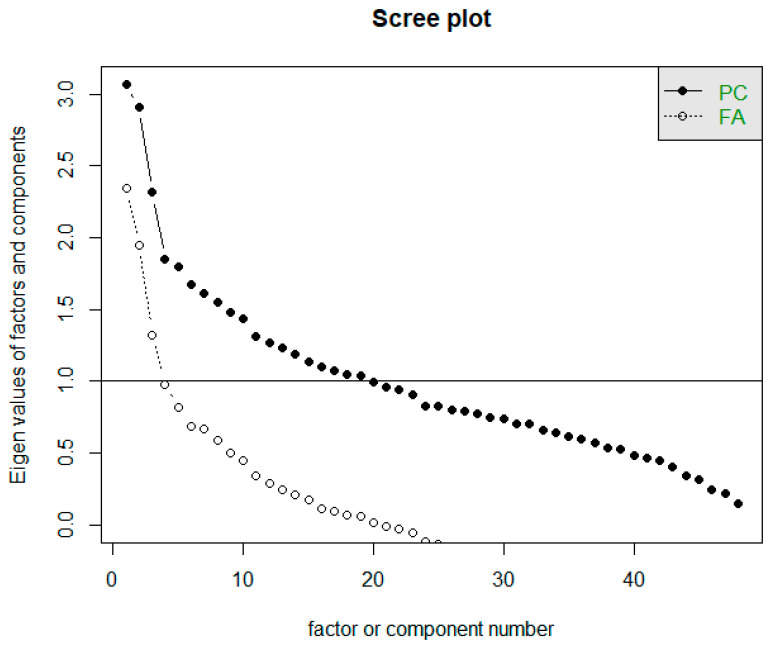
Representation of Scree plot to determine the number of factors.

**Figure 7 diagnostics-11-01284-f007:**
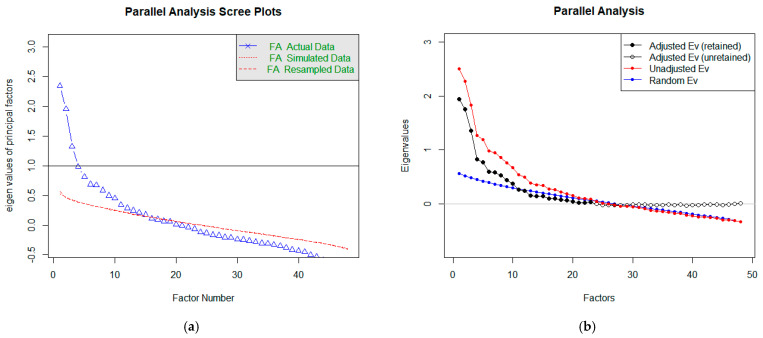
Representation of Parallel Analysis Scree plots to determine the number of factors.

**Figure 8 diagnostics-11-01284-f008:**
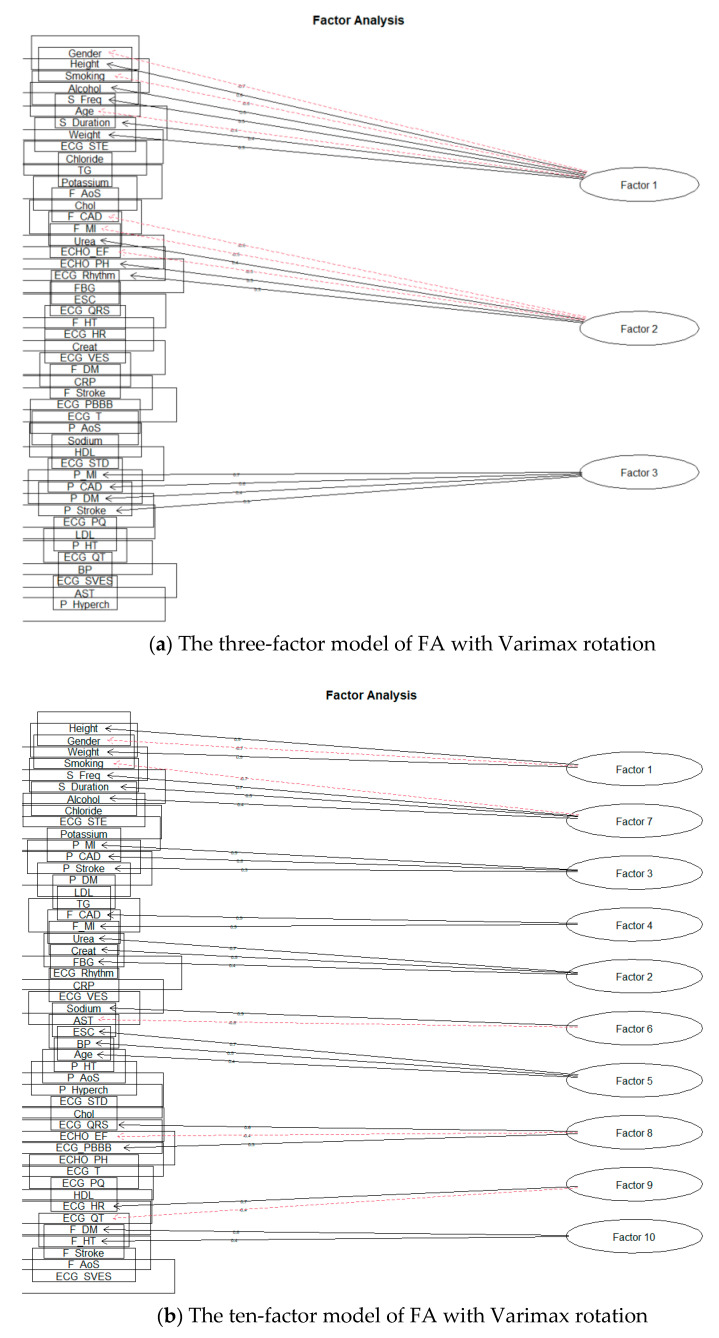
The solution of FA for Varimax rotation.

**Figure 9 diagnostics-11-01284-f009:**
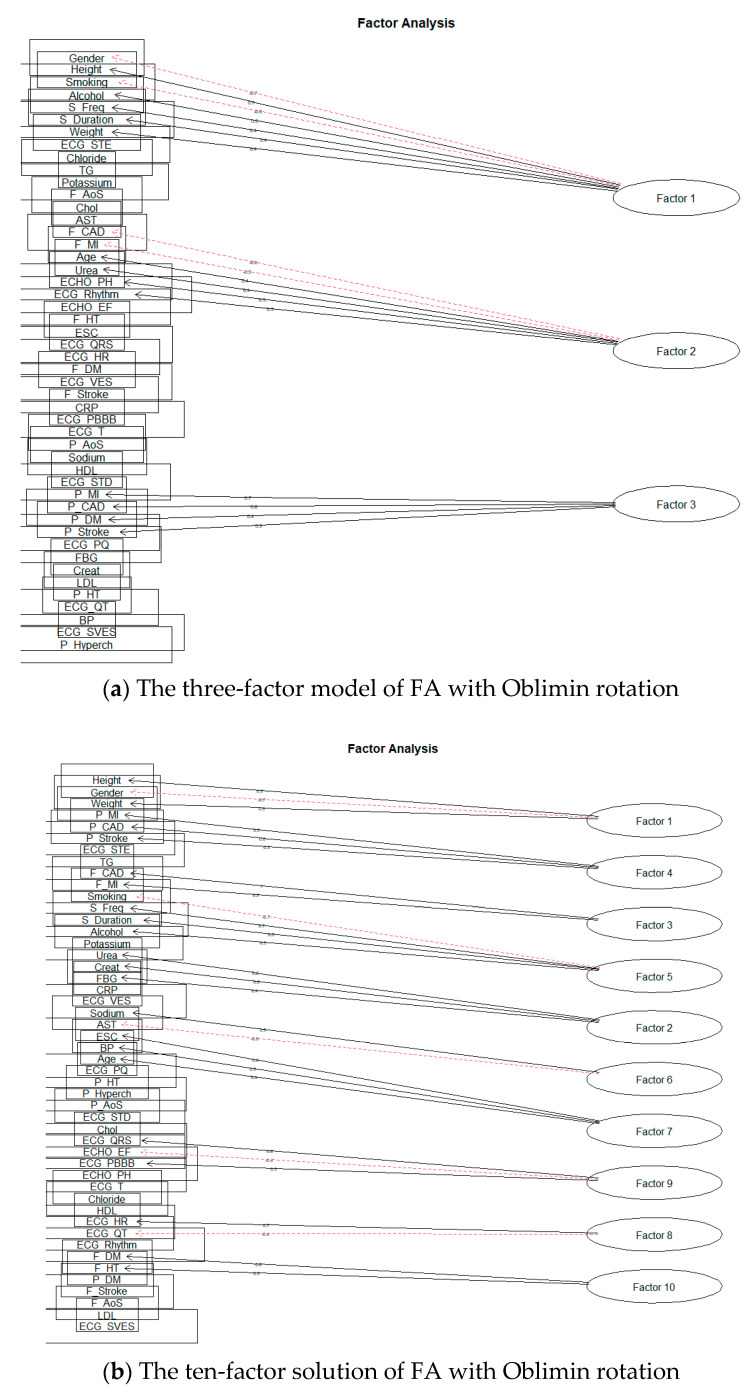
The solution of FA for Oblimin rotation.

**Figure 10 diagnostics-11-01284-f010:**
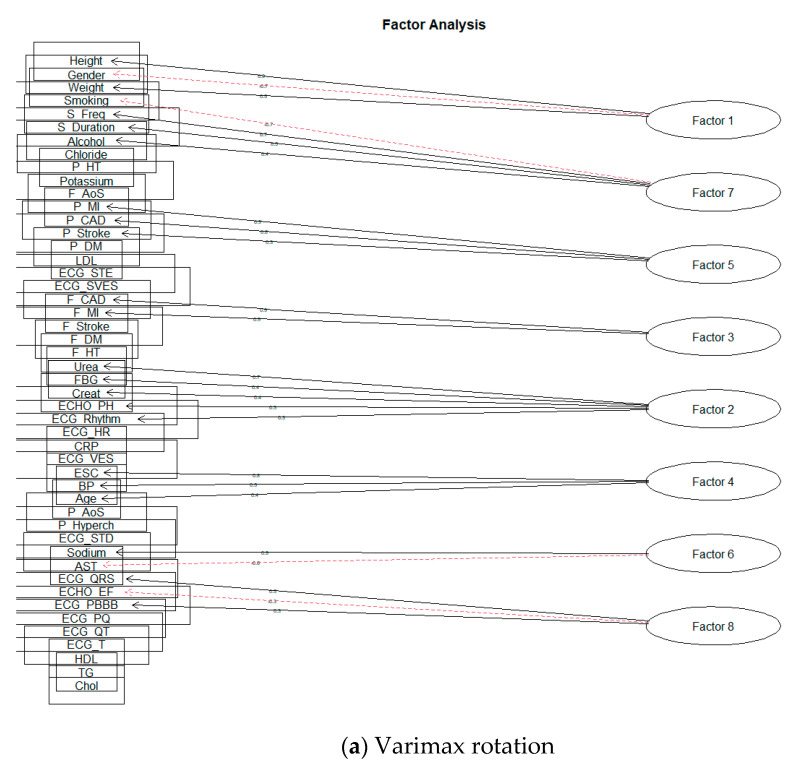
The eight-factor model of FA.

**Figure 11 diagnostics-11-01284-f011:**
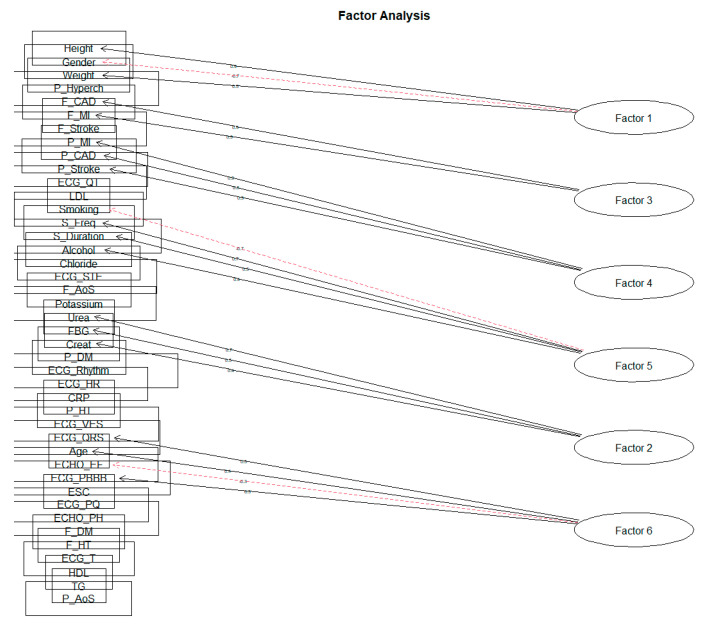
The six-factor solution for rotation Oblimin on the feature set.

**Table 1 diagnostics-11-01284-t001:** Overview of attributes.

Attribute	Type	Description	Values
Identifying attributes			
Id	numeric	according to a medical report	-
BN	text	encrypted birth number	-
Year	numeric	year of birth of the patient	min: 1930; mean: 1952.06; sd: 9.36; median: 1952; max: 1982; IQR: 13
Gender	binary	patient’s gender	0 (male): 460; 1 (female): 348
Symptomatic attributes			
P_CAD	binary	personal ischemic heart disease	FALSE: 478; TRUE: 329; NA: 1
P_Stroke	binary	personal stroke	FALSE: 703; TRUE: 104; NA: 1
P_MI	binary	personal infarct myocardium	FALSE: 591; TRUE: 216; NA: 1
P_Hyperch	binary	personal disease associated with high-level cholesterol	FALSE: 748; TRUE: 59; NA: 1
P_HT	binary	personal high blood pressure	FALSE: 264; TRUE: 543; NA: 1
P_DM	binary	personal type 2 diabetes	FALSE: 570; TRUE: 237; NA: 1
P_AoS	binary	personal aortic stenosis	FALSE: 738; TRUE: 69; NA: 1
F_CAD	binary	ischemic heart disease—occurrence in family	FALSE: 616; TRUE: 185; NA: 7
F_Stroke	binary	stroke—occurrence in family	FALSE: 702; TRUE: 104; NA: 7
F_MI	binary	infarct myocardium—occurrence in family	FALSE: 655; TRUE: 146; NA: 7
F_Hyperch	binary	disease associated with high-level cholesterol—occurrence in family	FALSE: 801; NA: 7
F_HT	binary	high blood pressure—occurrence in family	FALSE: 735; TRUE: 66; NA: 7
F_DM	binary	type 2 diabetes—occurrence in family	FALSE: 570; TRUE: 237; NA: 7
F_AoS	binary	aortic stenosis—occurrence in family	FALSE: 800; TRUE: 1; NA: 7
Smoking	categorical	type of smoker	1 (smoker): 100; 2 (ex-smoker): 140; 3 (non-smoker): 489; NA: 79
S_Duration	numeric	number of years of smoking	min: 0; mean: 2.79; sd: 8.43; median: 0; max: 60; IQR: 0; NA: 79
S_Freq	numeric	number of daily smoked cigarettes	min: 0; mean: 2.21; sd: 6.24; median: 0; max: 60; IQR: 0; NA: 79
Alcohol	binary	alcohol consumption	FALSE: 488; TRUE: 127, NA: 193
Weight	numeric	patient weight	min: 41; mean: 88; sd: 17.96; median: 88; max: 180; IQR: 20; NA: 27
Height	numeric	patient height	min: 45; mean: 165.74; sd: 11.65; median: 166; max: 193; IQR: 14; NA: 27
BMI	numeric	body mass index	min: 18.22; mean: 31.84; sd: 8.3; median: 31; max: 208.12; IQR: 7; NA: 27
BP	numeric	blood pressure	min: 12; mean: 167.91; sd: 510.18; median: 150; max: 14090; IQR: 30; NA: 54
Urea	numeric	blood urea level	min: 2.29; mean: 6.29; sd: 2.66; median: 5.69; max: 27.13; IQR: 2.42; NA: 26
Creat	numeric	blood creatinine level	min: 6.6; mean: 87.83; sd: 50.17; median: 80.4; max: 735.2; IQR: 25.53; NA: 14
AST	numeric	the level of enzyme secreted by the liver	min: 0.08; mean: 0.73; sd: 6.27; median: 0.38; max: 141; IQR: 0.16; NA: 303
Sodium	numeric	blood sodium level	min: 4.4; mean: 138.8; sd: 8.56; median: 139; max: 169.7; IQR: 3.75; NA: 7
Potassium	numeric	blood potassium level	min: 2.9; mean: 4.33; sd: 1.99; median: 4.26; max: 58.45; IQR: 0.70; NA: 18
Chol	numeric	total cholesterol	min: 0.82; mean: 5.78; sd: 22.03; median: 4.81; max: 614; IQR: 1.67; NA: 39
TG	numeric	level of triacylglycerols	min: 0.46; mean: 1.79; sd: 1.72; median: 1.41; max: 30.01; IQR: 0.91; NA: 59
HDL	numeric	high-density lipoprotein level	min: 0.51; mean: 1.47; sd: 3.93; median: 1.27; max: 108; IQR: 0.45; NA: 62
LDL	numeric	low-density lipoprotein level	min: 0.89; mean: 3.07; sd: 1.06; median: 2.94; max: 9.6; IQR: 1.42; NA: 84
CRP	numeric	C-reactive protein level	min: 0.1; mean: 5.86; sd: 10.88; median: 3.21; max: 130.4; IQR: 5.15; NA: 401
Chloride	numeric	blood chloride level	min: 91.8; mean: 103.5; sd: 3.38; median: 103.6; max: 111.2; IQR: 3.9; NA: 695
FBG	numeric	fibrinogen levels	min: 2.1; mean: 3.91; sd: 1.06; median: 3.7; max: 7.4; IQR: 1.25; NA: 661
HIV	binary	the presence of HIV	FALSE: 627; NA: 181
HBsAG	binary	the presence of an antigen evoking the presence of jaundice type B	FALSE: 626; NA: 182
ECG_HR	numeric	heart rate for minute	min: 45; mean: 69.54; sd: 12.23; median: 68; max: 130; IQR: 14; NA: 56
ECG_Rhythm	binary	type of heart rhythm	0 (SR): 704; 1(Fib): 60; NA: 44
ECG_PQ	numeric	the length of the interval from the beginning of the P wave to the beginning of the ventricular complex in milliseconds	min: 14; mean: 170.2; sd: 33.08; median: 160; max: 360; IQR: 30; NA: 170
ECG_QRS	numeric	heart ventricular depolarization time in milliseconds	min: 60; mean: 95.8; sd: 19.57; median: 90; max: 180; IQR: 20; NA: 117
ECG_QT	numeric	time from the beginning of the QRS to the end of the T wave in milliseconds	min: 40; mean: 386.4; sd: 43.57; median: 380; max: 518; IQR: 40; NA: 249
ECG_LBBB	binary	the presence of a blockage of the left Tawar arm	FALSE: 764; NA: 44
ECG_RBBB	binary	the presence of a blockage of the right Tawar arm	FALSE: 696; TRUE: 68; NA: 44
ECG_VES	binary	presence of ventricular extrasystoles	FALSE: 721; TRUE: 43; NA: 44
ECG_SVES	binary	presence of supraventricular (atrial) extrasystoles	FALSE: 739; TRUE: 25; NA: 44
ECG_STD	binary	the presence of depression in the ST segment	FALSE: 643; TRUE: 121; NA: 44
ECG_STE	binary	presence of elevations in the ST section	FALSE: 529; TRUE: 235; NA: 44
ECG_T	binary	ventricular myocardial repolarization	FALSE: 163; TRUE: 604; NA: 44
ECHO_EF	numeric	left ventricular ejection fraction	min: 15; mean: 52.72; sd: 9.93; median: 55; max: 75; IQR: 12; NA: 39
ECHO_PH	categorical	degree of pulmonary hypertension	0: 629; 1: 78; 2: 44; 3: 56; NA: 1
Resulting attributes			
Muscle_bridge	binary	the presence of a muscle bridge in one of the branches	FALSE: 802; TRUE: 5; NA: 1
ACS	numeric	percentage narrowing of the branch of Arteria coronaria sinistra	min: 0; mean: 6.35; sd: 19.44; median: 0; max: 100; IQR: 0; NA:1
RIA	numeric	percentage narrowing of the Ramus interventricularis anterior branch	min: 0; mean: 21.73; sd: 33.18; median: 0; max: 100; IQR: 0; NA: 1
RD1	numeric	percentage narrowing of branch RD1, part of RIA	min: 0; mean: 4.83; sd: 18.35; median: 0; max: 100; IQR: 0; NA: 1
RD2	numeric	percentage narrowing of branch RD2, part of RIA	min: 0; mean: 1.24; sd: 9.35; median: 0; max: 100; IQR: 0; NA: 1
RCX	numeric	percentage narrowing of the ramus circumflex artery branch	min: 0; mean: 17.18; sd: 30.86; median: 0; max: 100; IQR: 10; NA: 1
RIM	numeric	percentage narrowing of the RIM branch, part of the RCX	min: 0; mean: 2.55; sd: 12.69; median: 0; max: 100; IQR: 0; NA: 1
RMS1	numeric	percentage narrowing of the RMS1 branch, part of the RCX	min: 0; mean: 5.16; sd: 18.20; median: 0; max: 100; IQR: 0; NA: 1
RMS2	numeric	percentage narrowing of the RMS2 branch, part of the RCX	min: 0; mean: 2.08; sd: 12.09; median: 0; max: 100; IQR: 0; NA: 1
ACD	numeric	percentage narrowing of the Arteria coronaria dextra branch	min: 0; mean: 23.62; sd: 35.66; median: 0; max: 100; IQR: 50; NA: 1
RIP	numeric	percentage narrowing of the Ramus interventricularis posterior branch	min: 0; mean: 2.78; sd: 13.58; median: 0; max: 100; IQR: 0; NA: 1
Coron_result	categorical	the degree of severity of the finding	0: 310; 1: 67; 2: 36; 3: 103; 4: 164; 5: 126

Note: F—Family, P—Personal, sd—standard deviation, IQR—interquartile range.

**Table 2 diagnostics-11-01284-t002:** The Kaiser and Rice recommendation (1974) (Data from [20]).

Value of KMO	The Adequacy of the Observed Data Set
≥0.9	Excellent
<0.8; 0.9)	Commendable
<0.7; 0.8)	Moderately useful
<0.6; 0.7)	Average
<0.5; 0.6)	Weak
<0.5	Not enough

**Table 3 diagnostics-11-01284-t003:** An overview of the most important FA results for Varimax and Oblimin rotations.

Factors	Factor Loadings
**The three-factor solution for Varimax rotation**	
Factor 1	Gender; Height; Smoking; Alcohol; S_Freq; Age; S_Duration
Factor 2	F_CAD; F_MI; Urea; ECHO_EF; ECHO_PH; ECG_Rhythm
Factor 3	P_MI; P_CAD; P_DM; P_Stroke
**The ten-factor solution for Varimax rotation**	
Factor 1	Height; Gender; Weight
Factor 2	Urea; Creat; FBG
Factor 3	P_MI; P_CAD; P_Stroke
Factor 4	F_CAD; F_MI
Factor 5	ESC; BP; Age
Factor 6	Sodium; AST
Factor 7	Smoking; S_Freq; S_Duration; Alcohol
Factor 8	ECG_QRS; ECHO_EF; ECG_RBBB; ECHO_PH
Factor 9	ECG_HR; ECG_QT
Factor 10	F_DM; F_HT
**The three-factor solution for Oblimin rotation**	
Factor 1	Gender; Height; Smoking; Alcohol; S_Freq; S_Duration; Weight
Factor 2	F_CAD; F_MI; Age; Urea; ECHO_PH; ECG_Rhythm; ECHO_EF
Factor 3	P_MI; P_CAD; P_DM; P_Stroke
**The ten-factor solution for Oblimin rotation**	
Factor 1	Height; Gender; Weight
Factor 2	Urea; Creat; FBG
Factor 3	F_CAD; F_MI
Factor 4	P_MI; P_CAD; P_Stroke
Factor 5	Smoking; S_Freq; S_Duration; Alcohol
Factor 6	Sodium; AST
Factor 7	ESC; BP; Age
Factor 8	ECG_HR; ECG_QT
Factor 9	ECG_QRS; ECHO_EF; ECG_RBBB
Factor 10	F_DM; F_HT

**Table 4 diagnostics-11-01284-t004:** SS loading values for individual factors.

Factors	SS Loadings	Factors	SS Loading
Factor rotation ‘Varimax’		Factor rotation ‘Oblimin’	
**The three-factor solution**			
Factor 1	2.37	Factor 1	2.36
Factor 2	1.88	Factor 2	1.89
Factor 3	1.86	Factor 3	1.86
**The ten-factor solution**			
Factor 1	1.89	Factor 1	1.96
Factor 2	1.56	Factor 2	1.49
Factor 3	1.77	Factor 3	1.75
Factor 4	1.71	Factor 4	1.77
Factor 5	1.22	Factor 5	1.76
Factor 6	1.23	Factor 6	1.23
Factor 7	1.81	Factor 7	1.23
Factor 8	1.2	Factor 8	*0.98 **
Factor 9	*0.96 **	Factor 9	1.20
Factor 10	*0.93 **	Factor 10	*0.98 **

Note: * SS loading values below 1.

**Table 5 diagnostics-11-01284-t005:** SS loading values with factor loadings for individual solutions of FA.

Factors	SS Loadings	Factor Loadings
Factor rotation ‘Varimax’		
Factor 1	1.87	Height; Gender; Weight
Factor 2	1.65	Urea; FBG; Creat; ECHO_PH; ECG_Rhythm
Factor 3	1.76	F_CAD; F_MI
Factor 4	1.24	ESC; BP; Age
Factor 5	1.77	P_MI; P_CAD; P_Stroke
Factor 6	1.22	Sodium; AST
Factor 7	1.84	Smoking; S_Freq; S_Duration; Alcohol
Factor 8	1.10	ECG_QRS; ECHO_EF; ECG_RBBB
Factor rotation ‘Oblimin’		
Factor 1	1.97	Height; Gender; Weight
Factor 2	1.57	Urea; FBG; Creat
Factor 3	1.77	F_CAD; F_MI
Factor 4	1.25	ESC; BP; Age
Factor 5	1.77	P_MI; P_CAD; P_Stroke
Factor 6	1.23	Sodium; AST
Factor 7	1.78	Smoking; S_Freq; S_Duration; Alcohol
Factor 8	1.12	ECG_QRS; ECG_RBBB; ECHO_EF

**Table 6 diagnostics-11-01284-t006:** The cumulative variance of each n-factor solution.

Model	Cumulative Variance
Factor rotation Varimax	
The three-factor solution	13%
The eight-factor solution	26%
Factor rotation Oblimin	
The three-factor solution	13%
The eight-factor solution	26%

**Table 7 diagnostics-11-01284-t007:** SS loading values with factor loadings for the six-factors FA model with rotation Oblimin.

Factors	SS Loadings	Factor Loadings
Factor 1	1.97	Height; Gender; Weight
Factor 2	1.50	Urea; FBG; Creat
Factor 3	1.77	F_CAD; F_MI
Factor 4	1.75	P_MI; P_CAD; P_Stroke
Factor 5	1.76	Smoking; S_Freq; S_Duration; Alcohol
Factor 6	1.23	ECG_QRS; Age; ECHO_EF; ECKG_PBBB

**Table 8 diagnostics-11-01284-t008:** Values of factors loadings with the rate of variance for each factor.

	Factor 1	Factor 2	Factor 3	Factor 4	Factor 5	Factor 6
Height	0.890 *	−0.023	−0.006	−0.035	−0.036	−0.034
Gender	−0.702 *	−0.030	0.013	−0.041	−0.166	−0.156
Weight	0.510 *	0.193	−0.042	0.076	−0.082	−0.085
Urea	0.010	0.684 *	0.007	−0.044	−0.057	0.028
FBG	−0.012	0.463 *	−0.034	0.097	0.260	−0.033
Creat	0.103	0.436 *	0.027	0.006	−0.037	−0.046
F_CAD	−0.015	−0.002	0.912 *	−0.014	−0.009	0.003
F_MI	0.004	0.009	0.896 *	0.016	−0.009	0.008
P_MI	0.016	−0.017	0.027	0.869 *	0.007	0.009
P_CAD	−0.036	−0.006	−0.033	0.772 *	−0.015	−0.007
P_Stroke	−0.022	0.062	−0.044	0.329 *	−0.007	0.025
Smoking	−0.049	0.073	0.027	−0.057	−0.738 *	0.019
S_Freq	−0.029	0.038	0.014	−0.054	0.652 *	−0.079
S_Duration	−0.024	0.091	0.058	−0.060	0.536 *	0.075
Alcohol	0.201	−0.062	0.042	−0.079	0.361 *	0.039
ECG_QRS	0.153	−0.074	0.042	0.006	−0.086	0.530 *
Age	−0.372	0.207	−0.061	0.014	−0.092	0.414 *
ECHO_EF	−0.270	−0.162	0.016	−0.092	−0.033	−0.309 *
ECG_RBBB	0.012	−0.046	0.019	−0.041	−0.009	0.309 *
SS loading	1.965	1.499	1.771	1.753	1.755	1.230
Proportion Variance	0.047	0.036	0.042	0.042	0.042	0.029
Cumulative Variance	0.047	0.083	0.125	0.167	0.209	0.238

Note: * The highest achieved SS loading absolute value through the created factors.

## Data Availability

Due to patient confidentiality, it is impossible to provide access to the data examined in the present research.

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
