# Peer review of "Analysis of Risk Factors in Patients with Subclinical Atherosclerosis and Increased Cardiovascular Risk Using Factor Analysis"

_diagnostics, 2021, doi:10.3390/diagnostics11071284_

Round 1
Reviewer 1 Report
Brief summary
The aim of the study is to to to provide a less typical view of the issue of the cardiovascular risk factors.
Findings
The most significant cluster of factors identifies six factors: (1) Basic characteristics of the patient, (2) Renal parameters and fibrinogen, (3) Family predisposition to CVD, (4) Personal history of CVD, (5) Lifestyle of the patient, and (6) Echo and ECG examination results.
Strengths
The topic is of interest and the results are of relevance, even though it is a single center study.
Major issues
The introduction section should not describe the methodology of the study. The text after describing the aim should be deleted or incorporated to the methods section.
Th formatting of the discussion section should be rearranged. Please delete the numbers at the beginning of every paragraph, so that it doesn’t disrupt the reading flow. Please delete the bullet points from the results sections as well. The format of the text must be similar from the beginning to the end.
There is no limitations section.
There are some typographic errors. There are very long sentences. To improve readability, consider breaking this into multiple sentences. The authors are encouraged to proof-read thoroughly the text before resubmission. English must be excellent.
Reviewer 2 Report
The study findings are very promising but require some additional information to be included:
1) Please make the abstract more precisely because the current version looks pretty unorganized
2) Some blood parameters should be included, and use some references in the results sections.
3) In the discussion part, it is essential to describe some molecular-level information. For example, how the structure of the heart is changed during disease conditions.
Reviewer 3 Report
Pella and coll. In the present paper, they analyzed a database of hospitalized patients that underwent cardiological assessment in order to perform a factor analysis to assess clusters of characteristics potentially associated with cardiovascular risk and/or events.
The presentation is excessively long and somewhat confusing. Authors bring the reader among several irrelevant methodologic and technical aspects instead of providing a clear study design and presenting consolidated results. Conversely, the study population is not described nor the methods adopted for collecting primary data. Surprisingly, authors deserve a relevant portion of the space of this paper to the description of how they corrected imputation errors or filled missing data with a debatable method. By the way, the paper organization, far from the “classical” Introduction, Methods, Results, Discussion, Conclusions, does not help in understanding and recall what is truly relevant along with the presentation.
The same need for factor analysis in the clinical context of the present study is debatable. While some association between characteristics is simply obvious (see, for example, the association between BMI, height, and weight) other associations (e.g. obesity and insulin resistance) have been already well recognized and extensively analyzed in the past and recent literature. Accordingly, the authors conclude that the results of the present study are largely confirmatory. Anyway, it is extremely difficult to follow presented analyses since the excessive use of acronyms and the use of not-English terms.
Although factor analysis just identifies characteristics that cluster together without inferring their relationship with other variables (in this case, cardiovascular risk) authors along with this paper sometimes report their results as detection of relationships. This kind of conclusion is possible only by performing multiple regression analyses, controlling for confounding factors.
Figures are often unreadable and devoid of an explanatory legend.
Some reference seems not to be captured by the reference manager used by authors
Round 2
Reviewer 1 Report
All my queries have been addressed. I have no further comments regarding the manuscript.
Author Response
Thank you very much for the factual comments and remarks.
Reviewer 3 Report
The authors, in this revised manuscript, addressed most of the issues raised and improved the quality of the presentation.
Unfortunately, a simple Table 1 reporting characteristics of the study population is still missing. I would like that they report mean and standard deviation or, when appropriate, median and interquartile range or frequency, of age, gender, body height, weight, BMI, systolic and diastolic blood pressure, and so on. It remains difficult to correctly evaluate the results of advanced analyses (i.e. factor analysis) without seeing at least the distribution of the “primary” data.
I still suggest avoiding not-English terms in figures, tables, and text. The original name of a variable in the database is unimportant while the readability of the paper is severely impaired by the mix of terms not immediately understandable by the reader.
